# In-Season Cotton Yield Prediction with Scale-Aware Convolutional Neural Network Models and Unmanned Aerial Vehicle RGB Imagery

**DOI:** 10.3390/s24082432

**Published:** 2024-04-10

**Authors:** Haoyu Niu, Janvita Reddy Peddagudreddygari, Mahendra Bhandari, Juan A. Landivar, Craig W. Bednarz, Nick Duffield

**Affiliations:** 1Texas A&M Institute of Data Science, Texas A&M University, College Station, TX 77843, USA; duffieldng@tamu.edu; 2Department of Electrical & Computer Engineering, Texas A&M University, College Station, TX 77843, USA; janvita11@tamu.edu; 3AgriLife Research and Extension Center, Texas A&M University, Corpus Christi, TX 78406, USA; mahendra.bhandari@ag.tamu.edu (M.B.); juan.landivar@ag.tamu.edu (J.A.L.); 4Department of Agricultural Sciences, West Texas A&M University, Canyon, TX 79016, USA; cbednarz@wtamu.edu

**Keywords:** cotton, irrigation, UAV, convolutional neural networks, yield

## Abstract

In the pursuit of sustainable agriculture, efficient water management remains crucial, with growers relying on advanced techniques for informed decision-making. Cotton yield prediction, a critical aspect of agricultural planning, benefits from cutting-edge technologies. However, traditional methods often struggle to capture the nuanced complexities of crop health and growth. This study introduces a novel approach to cotton yield prediction, leveraging the synergy between Unmanned Aerial Vehicles (UAVs) and scale-aware convolutional neural networks (CNNs). The proposed model seeks to harness the spatiotemporal dynamics inherent in high-resolution UAV imagery to improve the accuracy of the cotton yield prediction. The CNN component adeptly extracts spatial features from UAV-derived imagery, capturing intricate details related to crop health and growth, modeling temporal dependencies, and facilitating the recognition of trends and patterns over time. Research experiments were carried out in a cotton field at the USDA-ARS Cropping Systems Research Laboratory (CSRL) in Lubbock, Texas, with three replications evaluating four irrigation treatments (rainfed, full irrigation, percent deficit of full irrigation, and time delay of full irrigation) on cotton yield. The prediction revealed that the proposed CNN regression models outperformed conventional CNN models, such as AlexNet, CNN-3D, CNN-LSTM, ResNet. The proposed CNN model showed state-of-art performance at different image scales, with the $R^{2}$ exceeding 0.9. At the cotton row level, the mean absolute error (MAE) and mean absolute percentage error (MAPE) were 3.08 pounds per row and 7.76%, respectively. At the cotton grid level, the MAE and MAPE were 0.05 pounds and 10%, respectively. This shows the proposed model’s adaptability to the dynamic interplay between spatial and temporal factors that affect cotton yield. The authors conclude that integrating UAV-derived imagery and CNN regression models is a potent strategy for advancing precision agriculture, providing growers with a powerful tool to optimize cultivation practices and enhance overall cotton productivity.

## 1. Introduction

Cotton holds significant importance in the global textile industry, accounting for approximately 25% of global fiber usage. The United States (US) stands out as a leading cotton exporter and ranks as the third-largest producer globally [1], with the northwest plain region of Texas, known as Texas High Plains (THP), serving as a vital contributor. Cotton is among one of the leading cash crops in Texas, with a total coverage of approximately 6 million acres [2]. The THP region alone contributes to about 25% and 65% of the US and Texas cotton production, respectively [3]. However, cotton cultivation in THP faces challenges, particularly in balancing water availability with yield optimization. While higher cotton yields are associated with increased water availability [4], water resource is currently insufficient to provide full irrigation in that region. The Lubbock area, for instance, is facing water limitation because of the significant decline of the water table in the Ogallala Aquifer [5]. Under such circumstances, accurate yield prediction will be very beneficial to the cotton production. Traditionally, remote sensing-based crop yield estimation with machine learning (ML) methods have been commonly used. For example, satellite remote sensing methods have been extensively utilized for crop yield estimation [6,7]. However, satellite images may suffer from occlusion by clouds, and their revisit cycle represents also a lack of flexibility. Although satellite remote sensing is valuable for large scale viewing of fields, its spatial resolution is still a concern for many precision agriculture applications [8].

In recent times, Unmanned Aerial Vehicles (UAVs) have emerged as valuable assets in diverse agricultural contexts, including the forecasting of yield [8,9], management of irrigation [10,11], and estimation of water stress [12,13]. Through the incorporation of lightweight sensors onto UAV platforms, it has become viable to capture imagery with exceptional spatial and temporal resolution at a minimal expense [14,15]. With ML regression models, such as support vector regression (SVR) [16] and random forest regression (RFR) methods [17], empirical regression models have been developed between crop yield and crop canopy features, such as vegetation indices [8,18,19]. For example, Ashapure et al. developed an ML framework for estimating cotton yield using multi-temporal remote sensing data collected from a UAV system [8]. Several types of crop features were derived to predict the yield, including the canopy cover, canopy height, canopy volume, normalized difference vegetation index (NDVI), excessive greenness index (ExG), etc. The model provided low residual values with predicted yield values close to the observed yield values ($R^{2}$ = 0.9). However, traditional ML methods may face limitations when applied to yield regression tasks in agriculture, particularly when using handcrafted features. First, traditional ML methods rely on handcrafted features, which may not capture the full complexity and richness of agricultural data. Features engineered by domain experts may overlook subtle patterns or interactions present in the data, leading to suboptimal performance. Second, traditional ML methods typically treat each input feature as independent, disregarding the spatial relationships present in agricultural data, such as UAV images of crop fields. These methods may overlook important spatial patterns and dependencies, leading to suboptimal performance in tasks requiring spatial understanding, such as crop yield prediction. Third, handcrafting features for agricultural datasets can be challenging and time-consuming, requiring domain expertise and experimentation. Moreover, manually engineered features may not fully capture the relevant information present in the data, leading to a loss of predictive power.

Expanding on previous investigations, this paper suggests employing UAV-captured RGB images solely for predicting cotton yield, utilizing convolutional neural networks (CNNs). CNN models are preferred over traditional machine learning algorithms for cotton yield prediction due to several key advantages. First, CNNs can automatically learn hierarchical representations of input data, extracting relevant features directly from the raw input images. Traditional machine learning algorithms often rely on handcrafted features, which may not capture the complex patterns present in image data as effectively as CNNs. Second, CNNs are well suited for processing spatial data, such as images, due to their ability to preserve spatial relationships between pixels. This is particularly important in cotton yield prediction, where features like crop health, vegetation density, and soil conditions can vary spatially within a field. Traditional machine learning algorithms may struggle to capture these spatial dependencies. Third, CNNs are inherently robust to noise and variability in input data, making them well suited for handling real-world challenges such as variations in lighting conditions, camera angles, and crop growth stages. Traditional machine learning algorithms may struggle to generalize to unseen variations in input data.

Due to their robust analytical capabilities, CNN models have found applications in various agricultural domains, including yield prediction [20,21,22], water stress analysis [23], and pest management [24]. For instance, Khaki et al. proposed a deep learning framework using CNNs and recurrent neural networks (RNNs) for crop yield prediction based on environmental data and management practices. The new model achieved a root-mean-square error (RMSE) of 9% and 8% of their respective average yields, substantially outperforming all other methods that were tested, such as random forest (RF) and deep fully connected neural networks (DFNNs) [20]. In [21], Sun et al. proposed a deep CNN-LSTM model for both end-of-season and in-season soybean yield prediction in Continental United States (CONUS) at the county level. The model was trained with crop growth variables and environment variables, which included weather data, MODIS Land Surface Temperature (LST) data, and MODIS Surface Reflectance (SR) data; historical soybean yield data were used as labels. The results of their experiment indicated that the prediction performance of their proposed CNN-LSTM model could outperform the pure CNN or LSTM model in both end-of-season and in-season.

Inspired by the landscape of CNN applications for yield regression in agriculture, in this article, the authors propose an innovative CNN framework for cotton yield prediction by employing KerasTuner [25] to optimize and search for the best CNN models. The current CNN regression models were built upon our previous CNN classification paper [26], where the architecture was relatively simple and fixed. In the previous paper, the CNN had a predetermined structure without much flexibility for customization. However, in our study, we have advanced upon this by implementing a more sophisticated approach. While CNNs have been widely utilized in agricultural yield prediction tasks, the use of KerasTuner introduces a novel approach to automatically search for the most effective CNN architecture for our specific application. This method allows us to efficiently explore a wide range of model architectures and hyperparameters, ultimately identifying the optimal configuration for predicting cotton yield accurately. Unlike some other frameworks that may have fixed architectures or require extensive manual tuning, KerasTuner offers a high level of abstraction and a user-friendly interface that simplifies the process of building, training, and tuning CNN models. Furthermore, our study stands out by addressing the challenge of dynamic input sizes. Unlike previous approaches that may have focused solely on a fixed input size, the proposed framework accommodates dynamic input sizes. This means that the proposed CNN models can seamlessly adapt to varying input resolutions, whether at the row level or grid level, where each grid represents only one square meter of area. This flexibility is crucial for accurately capturing spatial information and optimizing model performance across different scales of agricultural data. By integrating KerasTuner and accommodating dynamic input sizes, our approach offers a robust and versatile framework for yield regression in agriculture. It not only enhances the accuracy and efficiency of CNN-based yield prediction models but also enables seamless scalability and adaptability to diverse agricultural landscapes and datasets.

The objectives of this study are summarized as follows: (1) Evaluate the reliability of UAV-based RGB imagery in predicting cotton yields with different irrigation treatments. (2) Build a CNN model framework and demonstrate the performance of CNN, CNN-3D, Resnet, AlexNet, CNN-LSTM models on cotton yield prediction at the row and grid level. The major contributions of this article are as follows: (1) We devised a reproducible framework with customized CNN models for cotton yield prediction utilizing high-resolution RGB images obtained from UAVs. (2) We applied a KerasTuner method to dynamically search the best CNN model structure for cotton yield prediction. This approach offers a reliable and efficient solution for predicting yield in cotton crops.

The remainder of the manuscript is structured as follows: Section 2 provides a detailed overview of the materials and methods employed for in-season UAV-based cotton yield prediction. Following this, Section 3 offers an extensive analysis and discussion of cotton yield under various irrigation treatments, emphasizing the outcomes regarding the dependability of RGB imagery and the efficacy of CNN models. Lastly, Section 4 presents concluding remarks that encapsulate the main insights and implications of the research.

## 2. Materials and Methods

### 2.1. The Study Site and Yield Data Collection

The research was carried out in an experimental cotton field situated at the USDA-ARS Cropping Systems Research Laboratory (CSRL) in Lubbock, Texas, USA ($33.69^{{}^{\circ}}$ N, $101.82^{{}^{\circ}}$ W). Cotton planting took place on 3 May 2022, using NG 4098 B3XF. The cotton field was partitioned into 12 drip zones (Figure 1), each replicated three times, to evaluate four distinctive irrigation treatments: “rainfed”, “full irrigation”, “percent deficit of full irrigation”, and “time delay of full irrigation”. Each drip zone comprised eight rows, and cotton fiber was mechanically harvested from individual rows within each drip zone, resulting in a dataset of 96 rows of cotton yield data. Each row spanned 200 feet, accommodating approximately 150 cotton plants, with a 40-inch spacing between rows [26].

Under the “rainfed” treatment, soil moisture conditions before planting determined irrigation, either through natural rainfall or pre-sowing soil irrigation on a predetermined date. No additional water was applied during the growing season under this treatment. In the “full irrigation” treatment, irrigation events were initiated based on accumulated stress time derived from canopy temperature, with the volume adjusted to replenish soil moisture deficits. Irrigation was administered as single events whenever necessary to sustain optimal soil moisture levels. The “percent deficit of full irrigation” triggered irrigation at a predetermined fraction (25%), of the volume used in the “full irrigation” treatment, maintaining the same frequency but with reduced volume. In the “time delay of full irrigation" treatment, irrigation events occurred alternatively with the “full irrigation” signal, replenishing the soil profile to full capacity, resulting in approximately 50% less water application compared to the “full irrigation” treatment. However, this treatment subjected the cotton to longer periods of water stress.

### 2.2. Description of the UAV and RGB Image Processing

A UAV platform, specifically, a DJI Phantom 4, was used to capture high-resolution RGB images from an altitude of 90 m, producing images sized at 4096 × 2160 pixels. The onboard camera boasts a 1-inch 20-megapixel (MP) complementary metal–oxide–semiconductor (CMOS) sensor with a mechanical shutter, thus avoiding rolling shutter distortion. This sophisticated sensor, along with robust processing capabilities, captures fine details crucial for subsequent advanced post-production analysis. Flight missions occurred biweekly throughout the cotton growing season, from May to October in 2022. Following each mission, the UAV RGB images underwent seamless stitching to produce orthomosaic images using Metashape (Agisoft LLC, St. Petersburg, Russia). Because of the reliable image quality, the data from the following four dates were selected, 18 August, 2 September, 9 September, and 20 September in 2022 [26].

The study aimed to predict cotton yield using advanced CNN models. To effectively analyze the robustness of the following proposed CNN models. The authors trained the CNN models at two different image scales, at the row and grid level. For the 96 row cotton images, as mentioned earlier, yield data were collected from each row. The original image size for the row cotton image was 1792 × 32 × 3. To better fit the row cotton images into CNN models, each row image was resized to 256 × 224 × 3. For the grid cotton images, the authors split the large-scale UAV cotton image into a grid scale with ArcGIS Pro, which created 5376 images for each sampling date (Figure 2 is a demonstration of the generated dataset [26]). The new image size was 32 × 32 × 3. The corresponding irrigation treatment was also added at the bottom of each image in Figure 2 for demonstration.

### 2.3. Convolutional Neural Networks

In this paper, the authors developed a comprehensive framework for cotton yield prediction, leveraging the power of deep learning techniques. The framework comprised five customized CNN models tailored specifically for cotton yield prediction. In the following section, we delve into an exploration of the CNN model, CNN-LSTM, ResNet, CNN-3D, and AlexNet, which offer unique advantages in various prediction tasks. For more detailed information and specific instructions on replicating the models discussed, please refer to the “Reproducibility” section located at the end of this paper.

#### 2.3.1. A Customized Convolutional Neural Network

As mentioned earlier, authors divided the large-scale UAV cotton image into row and grid level. To create a training and testing dataset, the UAV dataset was split into training (80%) and testing (20%) sets. For example, to prepare the grid-level cotton images for input into the CNN models, all the images were resized to a dimension of 32 × 32 × 12. The implementation of the CNN model relied on the TensorFlow 2.0 framework [27]. An illustration depicting the architecture of the CNN model is provided in Figure 3. This model predominantly comprised layers such as Conv2D and MaxPooling2D, where each layer yielded a 3D tensor output representing dimensions of height, width, and channels. With increasing depth of the network, the width and height dimensions tended to diminish. The number of output channels for each Conv2D layer was determined by the first specified argument. Subsequently, the output tensor from the convolutional base was directed to Dense layers for yield prediction in the form of a regression model. The CNN model utilized in this research can be readily replicated for validation purposes.

#### 2.3.2. Convolutional Neural Networks with Long Short-Term Memory (CNN-LSTM) Networks

The incorporation of long short-term memory (LSTM) networks in our model stems from the key idea that crop yield is influenced by temporal variations in crop growth across different months [28]. Unlike a traditional 2D CNN, where images from various months are stacked along the same axis, we introduced a fourth dimension to consider the temporal factor explicitly. The unique recursive structure of LSTM incorporates a sophisticated gating mechanism, which effectively controls the flow of information in and out of cells. This gating mechanism not only regulates the entry and exit of data but also facilitates the processing of sequential information and time series data. The intricate design of LSTM enables it to effectively capture and retain long-range dependencies in sequential data, distinguishing it from traditional RNNs [29]. Our study leveraged a CNN-LSTM architecture to investigate the hypothesis that crop yield depends on both temporal and spatial features. The CNN component is good at capturing spatial and hierarchical features inherent in crop images whereas the LSTM component proves advantageous in identifying temporal patterns crucial for understanding crop yield variations throughout their growth cycle.

The architecture of our model involved hyperparameter tuning using the KerasTuner to optimize the CNN and LSTM layers, kernel size, learning rate, optimizer, and dropout rates. The TimeDistributed CNN section applied convolutional layers along the temporal dimension of the input. For example, for a row cotton image, the input given to the model was 4 × 256 × 224 × 3, where 4 was the number of timestamps. The number of CNN layers was flexibly determined by varying the depth from 1 to 5 layers, each consisting of a Conv2D operation with a ReLU activation function and BatchNormalization. MaxPooling was applied after the initial CNN layer, enhancing the model’s ability to discern spatial features. The kernel size, a critical hyperparameter, was sampled from a uniform distribution between 2 and 5. Subsequently, the output from the CNN section was flattened using TimeDistributedFlatten, providing the input for the subsequent LSTM layers. The number of LSTM layers was also tuned to determine the optimal depth. Each LSTM layer exhibited a diminishing number of units, and dropout rates varying from 0.3 to 0.7 were applied for regularization purposes. The optimization further extended to the choice of optimizer, offering alternatives such as Adam, SGD, RMSprop, and Adagrad. The learning rate, as with previous models, was sampled from a logarithmic scale between 1$\times 10^{-4}$ and 1$\times 10^{-2}$. The model was compiled using the specified optimizer, employing the mean squared error as the loss function and the MAE as the evaluation metric.

#### 2.3.3. Residual Network (ResNet)

We developed a customized ResNet network for yield prediction tasks. The incorporation of ResNet’s concept of residual connections serves to enhance the depth of the network and mitigate the degradation problem when the accuracy gets saturated [30]. This innovation involves implementing skip connections between layers, allowing the model to learn the disparities between input and output across consecutive layers. This design choice mitigates the vanishing gradient problem during backpropagation, ensuring more effective and stable training. The convolutional layer comprises a Conv2D filter, ReLU as the activation function, max pooling, and batch normalization. Identity blocks are stacked sequentially, each composed of two convolutional layers, followed by skip connections referencing the outputs of the previous two layers. This architectural configuration aims to capture essential features in the data, enhancing the model’s predictive capabilities.

We also conducted a hyperparameter tuning process on the ResNet model using the Keras framework. A maximum of 10 tuning trials were executed to minimize the MAE of the validation loss. This exhaustive tuning process aimed to identify the most effective combination of hyperparameters, aligning with our primary objective of accurate yield prediction. The kernel size parameter, determining the dimensions of the convolutional filters, was sampled from a uniform distribution ranging between 2 and 5. The number of identity blocks, crucial components in our architecture, underwent variation from a simple configuration with one layer to a more intricate structure with four layers, allowing us to explore diverse model complexities. The number of filters varied in multiples of 32 as the depth continued to increase. To enhance learning dynamics, the learning rate was tuned by sampling a logarithmic scale ranging from 1$\times 10^{-4}$ to 1$\times 10^{-2}$. This ensured a balanced exploration of learning rates that were essential for convergence and model performance. We considered various optimizers like Adam, SGD, RMSprop, and Adagrad. This comprehensive exploration aimed to identify the most suitable optimizer for our yield prediction tasks.

#### 2.3.4. Three-Dimensional Convolutional Neural Networks (CNN-3Ds)

Interpreting spectral data poses challenges, particularly in the analysis and comparison of multiple samples over extended periods [31]. Existing classification methods, such as 1D-CNN (spectral feature-based) and 2D-CNN (spatial feature-based), face limitations because they do not effectively integrate spatial and spectral features. In contrast, a 3D-CNN or CNN-3D excels at extracting spatial–spectral features from volumetric data. Its ability to incorporate the spectral dimension alongside spatial dimensions allows it to model complex spatiotemporal representations. To address this, we also designed a CNN-3D architecture for the prediction of cotton yield.

The model was designed to handle 3D volumes with dimensions (256, 224, number of time stamps, 3). The convolutional layers were dynamically hypertuned by constructing flexible options for the number of layers (2 to 6) and kernel size (2 to 4). Three-dimensional filters/kernels slid over the input volume, capturing both local patterns and features [32]. A ReLU activation was applied to the convolutional layers, and the spatial downsampling was achieved through MaxPooling3D after the initial two layers. The architecture incorporated batch normalization after convolutional layers for enhanced training stability. A flatten layer preceded a two-layered Multi-Layer Perceptron (MLP) with ReLU activation and dropouts, serving as regularization. The output layer, indicative of a regression problem, consisted of a single-unit dense layer with ReLU activation. Hyperparameter choices, including the optimizer (Adam, SGD, RMSprop, or Adagrad) and a tunable learning rate (in the range (1$\times 10^{-4}$, 1$\times 10^{-2}$)), were determined during model configuration. The best set of hyperparameters was selected by comparing the mean absolute error on the validation data.

#### 2.3.5. AlexNet

The AlexNet architecture was also used to predict crop yield, utilizing input data with dimensions (256, 224, 12). This architecture extracts hierarchical features through a sequence of convolutional, normalization, activation, and pooling layers, with dropout incorporated after the flattened layers to mitigate overfitting [33]. Comprising 5 convolutional layers and 3 fully connected layers, each convolutional layer involves a convolution operation, followed by max pooling for spatial dimension reduction, and ReLU activation for non-linearity and a faster training process [34]. The flattened layer transforms the output from convolutional layers into a 1-dimensional array, preparing it for subsequent fully connected layers. The first dense layer consisted of 400 neurons with a ReLU activation function. Dropout layers followed the first and second dense layers to enhance generalization. The final dense layer, with 1 neuron, indicated a regression objective. To optimize the AlexNet model’s efficiency, a parameter tuning strategy was implemented using KerasTuner. The goal was to maximize predictive performance by discovering the optimal combination of hyperparameters. Hyperparameters, such as kernel size, optimizer choice, and learning rate, were defined as tunable parameters through KerasTuner. Kernel sizes for each convolution layer were explored within the range of 2 to 5, and optimizer choices encompassed Adam, SGD, RMSprop, and Adagrad. Learning rates for the selected optimizers were sampled logarithmically between 1$\times 10^{-4}$ and 1$\times 10^{-2}$.

### 2.4. CNN Regression Models’ Evaluation Metrics

When evaluating the performance of CNN regression models, several metrics are utilized to assess their accuracy and predictive capabilities. The mean absolute error (MAE) serves as a fundamental metric, representing the average absolute difference between the predicted and observed values. A lower MAE indicates better model performance, as it signifies smaller deviations between predictions and actual outcomes.
(1)$$\mathrm{MAE}=\frac{1}{n}\sum_{i=1}^{n}\left|y_{i}-{\hat{y}}_{i}\right|,$$
where $y_{i}$ represents the actual observed value for the $i_{th}$ data point, $\hat{y_{i}}$ is the predicted value for the $i_{th}$ data point, *n* stands for the total number of data points.

The mean absolute percentage error (MAPE) supplements MAE by expressing the average percentage deviation between predicted and observed values, providing insights into the relative magnitude of errors.
(2)$$\mathrm{MAPE}=\frac{1}{n}\sum_{i=1}^{n}\left|\frac{y_{i}-{\hat{y}}_{i}}{y_{i}}\right|\times 100\%.$$

Furthermore, the coefficient of determination, often denoted as $R^{2}$, offers a measure of the proportion of variance in the dependent variable that is predictable from the independent variables. A higher $R^{2}$ value indicates a better fit of the model to the data, with values closer to 1 signifying a stronger predictive power.
(3)$$R^{2}=1-\frac{\sum_{i=1}^{n}{\left(y_{i}-{\hat{y}}_{i}\right)}^{2}}{\sum_{i=1}^{n}{\left(y_{i}-\bar{y}\right)}^{2}},$$
where $\bar{y}$ represents the mean of the observed values across all data points. Together, these evaluation metrics offer a comprehensive assessment of CNN regression models, enabling researchers to gauge their accuracy, reliability, and generalization capabilities in predicting continuous outcomes.

## 3. Results and Discussion

### 3.1. Exploratory Cotton Yield Analysis

The irrigation treatment played a vital role in cotton growth, and its impact on cotton yield was evaluated based on the cotton harvest weight (Figure 4). To perform exploratory data analysis (EDA) for the provided row cotton yield data, the authors first observed that the data were structured in rows representing different conditions: “rainfed”, “percent deficit”, “time delay”, and “fully irrigated”. Each condition had corresponding mean yield values (in pounds), standard deviation values (in pounds), and counts of observations (Table 1). Significant variations could be observed in the mean cotton yield under different conditions. The “fully irrigated” cotton had the highest mean cotton yield at 58.68 pounds, with a standard deviation of 9.31 pounds, indicating considerable variability within the data. “Time delay” conditions exhibited the next highest mean yield of 38.57 pounds with a standard deviation of 2.89 pounds, suggesting a relatively lower variability compared to fully irrigated conditions. “Rainfed” conditions had a mean yield of 19.92 pounds and a standard deviation of 2.66 pounds, showing the lowest mean yield among the conditions with moderate variability. Notably, the “percent deficit” cotton yielded a mean cotton yield of 34.26 pounds with a standard deviation of 5.72 pounds, indicating intermediate levels of yield and variability.

To test the performance of the CNN models at a smaller field scale (grid level), the authors divided the large-scale UAV cotton image into smaller scales, resulting in a total of 5376 images. Generating grid cotton yield data involves creating a spatially distributed dataset that represents cotton yield across the field. To realize it, a linear regression model was employed to elucidate the relationship between blue band reflectance data derived from UAV RGB images and cotton yield obtained from row cotton yield measurements. Then, a quantitative regression model between the blue band reflectance from the cotton canopy cover and grid cotton yield was established (Figure 5).

The grid yield dataset offered a robust depiction of cotton yield dynamics within various irrigation treatments. Each treatment configuration was characterized by its mean yield, standard deviation, and observation count, providing a comprehensive understanding of cotton productivity across distinct agricultural regimes (Table 1). Noteworthy was the discernible variation in mean yield levels and standard deviations across treatments, with “fully irrigated” conditions exhibiting the highest mean yield of 1.02 pounds and the lowest standard deviation of 0.12 pounds, indicative of optimized agricultural management practices. Conversely, “rainfed” conditions manifested the lowest mean yield at 0.36 pounds, accompanied by a standard deviation of 0.14 pounds, implying heightened yield variability and potential vulnerability to environmental stressors (Figure 6).

The emergence of a few negative yield data points within the context of a linear regression model between blue band and yield variables can stem from various sources inherent to the data and modeling process. Negative yield predictions may arise due to extrapolation beyond the observed range of the data, particularly when the linear regression model attempts to estimate yield values that fall outside the bounds of the training dataset. Additionally, statistical noise, outliers, and violations of underlying assumptions regarding the relationship between predictor and response variables can contribute to negative predictions. While negative yield values may seem counterintuitive and potentially erroneous, retaining them during the training of CNN models can offer several advantages. First, negative yield values provide valuable diversity and richness to the training dataset, enabling CNN models to learn and generalize across a wider spectrum of potential scenarios and data distributions. Second, incorporating negative yield data points into the training process helps foster robustness and adaptability within CNN models, facilitating the extraction of intricate patterns and relationships present within the data. Moreover, by systematically addressing and accounting for negative yield predictions during CNN model training, researchers and practitioners can develop more resilient and insightful predictive models that better reflect the complexities of real-world agricultural systems. Thus, while negative yield data may pose initial challenges, leveraging them judiciously in CNN model training can yield substantial benefits in improving model performance and predictive accuracy. These insights elucidate the intricate interplay between environmental factors and crop performance, offering critical foundations for precision agriculture strategies and sustainable agricultural development initiatives.

### 3.2. The Performance of CNN Models at the Row Level

As mentioned previously, the study utilized a total of 96 cotton row images for each sampling date: 18 August, 2 September, 9 September, and 20 September, all in 2022. To concatenate images along the third dimension, the authors essentially stacked them together depth-wise. Each day contributed 96 UAV RGB images, each with dimensions 32 × 32 × 3 pixels. Since each image had three color channels (red, green, and blue), the resulting concatenated dataset had a depth of 12, representing the three color channels multiplied by the four sampling days. Thus, after concatenation, the new dataset had a size of 96 images with dimensions 32 × 32 × 12. This concatenated dataset effectively combined the information from all four sampling days into a single dataset, enabling a comprehensive analysis and processing of the collected image data across multiple time points.

Then, the concatenated dataset was split into a training set (80%) and a testing set (20%). The results for cotton yield prediction models indicated varying performance metrics across different CNN architectures (Table 2 and Figure 7). Starting with the MAE, it was observed that the customized CNN model achieved the lowest MAE of 3.08 lb, indicating the closest average prediction to the actual yield values. Following closely was the AlexNet model with an MAE of 4.84 lb, indicating slightly higher prediction errors compared to the customized CNN. However, both ResNet and CNN-3D models exhibited higher MAE values of 5.44 lb and 5.25 lb, respectively, suggesting relatively larger prediction errors compared to the CNN and AlexNet models. The CNN-LSTM model, on the other hand, demonstrated a significantly higher MAE of 11.97 lb, indicating considerable deviation between predicted and observed yield values.

Moving on to the MAPE, which measures the percentage difference between predicted and observed values, we observed a similar trend. The customized CNN model achieved the lowest MAPE of 7.76%, indicating the smallest average percentage deviation from the actual yield values. The AlexNet and ResNet models followed, with MAPE values of 14.4% and 13.53%, respectively, indicating relatively higher percentage errors compared to the customized CNN model. The CNN-3D model showed a slightly lower MAPE of 12.08%, suggesting better performance than AlexNet and ResNet but still higher than the customized CNN. However, the CNN-LSTM model exhibited a significantly higher MAPE of 35.07%, indicating substantial percentage deviation between predicted and observed values, which may be attributed to its architecture’s limitations in capturing temporal dependencies effectively. When dealing with a smaller dataset, such as the one with only 96 row images, the CNN-LSTM model may struggle to effectively capture the temporal dependencies present in the data, leading to suboptimal performance despite our hyperparameter tuning efforts with KerasTuner.

Finally, examining the coefficient of determination ($R^{2}$), which indicates the proportion of the variance in the dependent variable that is predictable from the independent variable(s), one can observe consistent trends (Figure 8). The proposed CNN model achieved the highest $R^{2}$ value of 0.93, indicating a strong correlation between predicted and observed yield values. The AlexNet, ResNet, and CNN-3D models exhibited progressively lower $R^{2}$ values of 0.84, 0.80, and 0.76, respectively, suggesting decreasing predictive power compared to the customized CNN model. Notably, the CNN-LSTM model demonstrated a negative $R^{2}$ value of −0.03, indicating poor model performance and a possibly inadequate capture of temporal dynamics, leading to predictions that were worse than simply using the mean of the observed values.

### 3.3. The Performance of CNN Models at the Grid Level

As mentioned earlier in the article, the row cotton images were further split at the grid level. Each grid image spanned a one-square-meter area. Through a systematic concatenation process along the third dimension, the images were effectively consolidated into a single dataset. Given that each image inherently encompassed three color channels—red, green, and blue—the resultant concatenated dataset manifested a depth of 12, representing the amalgamation of color channels across the four sampling days. As a consequence, the unified dataset comprised 5376 images, each having dimensions of 32 × 32 × 12 pixels. The corresponding yield ground truth was derived with the linear regression model described in Section 3.1.

For the MAE, the proposed CNN and AlexNet models showcased competitive performance with MAEs of 0.05 lb and 0.05 lb, respectively, outperforming ResNet and CNN-3D with higher MAEs of 0.09 lb and 0.09 lb, respectively (Table 3 and Figure 9). However, CNN-LSTM achieved comparable results to the proposed CNN with an MAE of 0.05 lb. In terms of MAPE, both the proposed CNN and AlexNet models exhibited similar performance, with MAPEs of 10.00% and 10.08%, respectively, outperforming ResNet and CNN-3D with higher MAPEs of 15.61% and 14.17%. Again, CNN-LSTM demonstrated competitive results with an MAPE of 10.46%, similar to that of the proposed CNN. Concerning $R^{2}$, both the proposed CNN and AlexNet models demonstrated superior predictive power with $R^{2}$ values of 0.95 and 0.96, respectively, surpassing ResNet and CNN-3D with lower $R^{2}$ values of 0.84 and 0.85 (Figure 10). Similarly, CNN-LSTM achieved a high $R^{2}$ of 0.95, comparable to that of the proposed CNN, indicating strong predictive relationships for both models.

The findings suggest that CNN models exhibit promising capabilities in predicting cotton yield across different field scales. Particularly noteworthy is the performance of the proposed CNN architecture, which demonstrated exceptional predictive accuracy across various field scale images without necessitating extensive hyperparameter tuning efforts. This highlights the robustness and versatility of the proposed CNN model, which can effectively generalize its learned representations to different field levels. Moreover, the comparative analysis revealed that all CNN architectures evaluated in the study exhibited commendable performance in predicting cotton yield, indicating the presence of a robust and reliable framework for cotton yield prediction using CNN models. This collective success underscores the potential of CNN-based approaches in agricultural applications, offering stakeholders a diverse array of models to choose from based on specific requirements and preferences.

## 4. Conclusions

As agriculture strives for sustainability, effective water management remains paramount, prompting growers to seek sophisticated methodologies for informed decision-making. Within this context, the prediction of cotton yield emerges as a critical aspect of agricultural planning, necessitating the utilization of cutting-edge technologies. However, conventional methods often falter in encapsulating the nuanced intricacies of crop health and growth dynamics. This study introduced a pioneering approach to cotton yield prediction, harnessing the synergy between UAVs and scale-aware CNNs. Our novel model endeavored to exploit the inherent spatiotemporal dynamics present in high-resolution UAV imagery to enhance the accuracy of cotton yield prediction. The research findings demonstrated the superiority of the proposed CNN regression models over conventional CNN architectures such as AlexNet, CNN-3D, CNN-LSTM, and ResNet. We applied a KerasTuner method to dynamically search the best CNN model structure for cotton yield prediction. Remarkably, our proposed CNN model showcased state-of-the-art performance across various image scales, boasting an impressive R-squared value exceeding 0.9. At the macro-level of individual cotton rows, the MAE and MAPE stood at 3.08 pounds per row and 7.76%, respectively. At the micro-level of the cotton grid, the MAE and MAPE were recorded as 0.05 pounds and 10%, respectively.

These results underscore the adaptability of our approach to the dynamic interplay between spatial and temporal factors influencing cotton yield. In summation, the integration of UAV-derived imagery and CNN regression models emerges as a potent strategy for advancing precision agriculture. This integration equips growers with a robust toolkit to optimize cultivation practices, fostering enhanced cotton productivity and sustainable agricultural outcomes. For future research endeavors, the authors plan to incorporate the latest cotton yield data from the year 2023. By integrating this updated dataset, our proposed CNN regression model can be subjected to testing with multiple years of data. This expansion in the temporal scope of the study will allow for a more comprehensive assessment of the model’s performance and robustness across varying growing seasons and environmental conditions.

## 5. Research Reproducibility

All the research results in this article can be reproduced. The code is available in the author’s Github: https://github.com/hniu-tamu/In-season-cotton-yield-prediction-with-CNN-models, accessed on 13 March 2024.

## Figures and Tables

**Figure 1 sensors-24-02432-f001:**
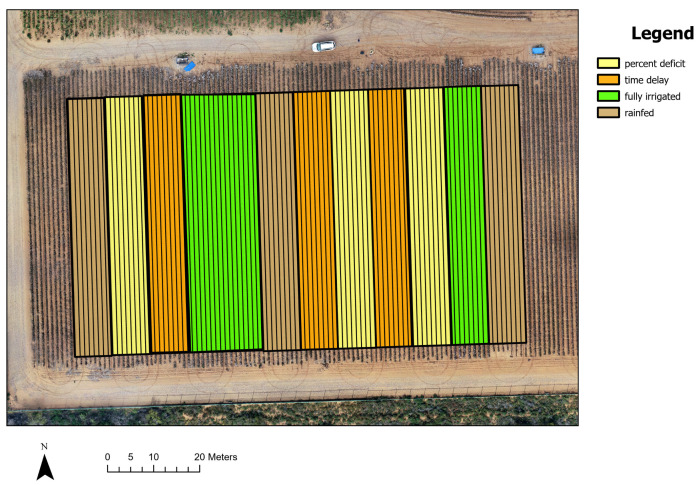
The experimental cotton field.

**Figure 2 sensors-24-02432-f002:**
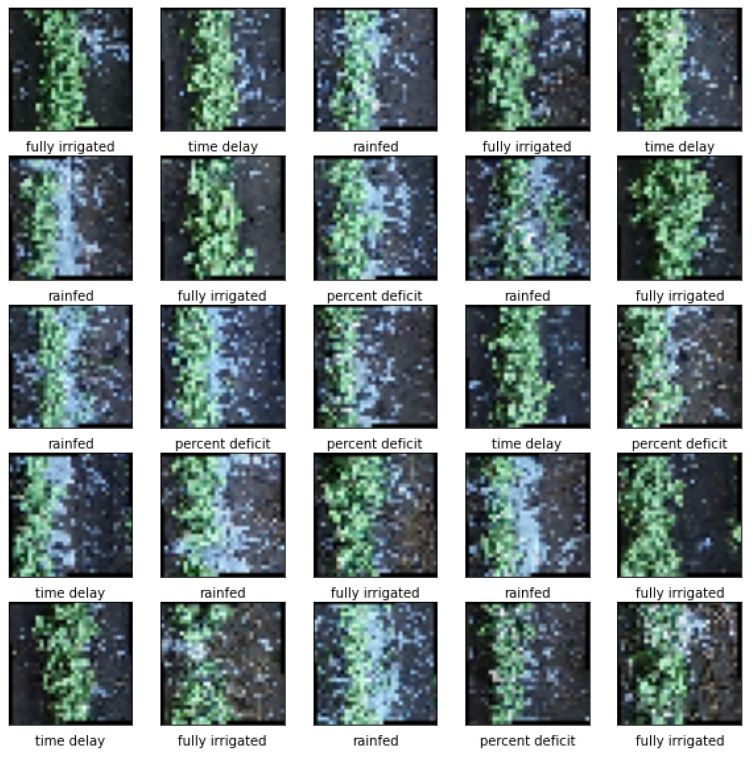
To predict the cotton yield with CNN models at the grid level, the authors first split the large scale of UAV cotton image into smaller scales with ArcGIS Pro, which created 5376 images for each sampling date.

**Figure 3 sensors-24-02432-f003:**
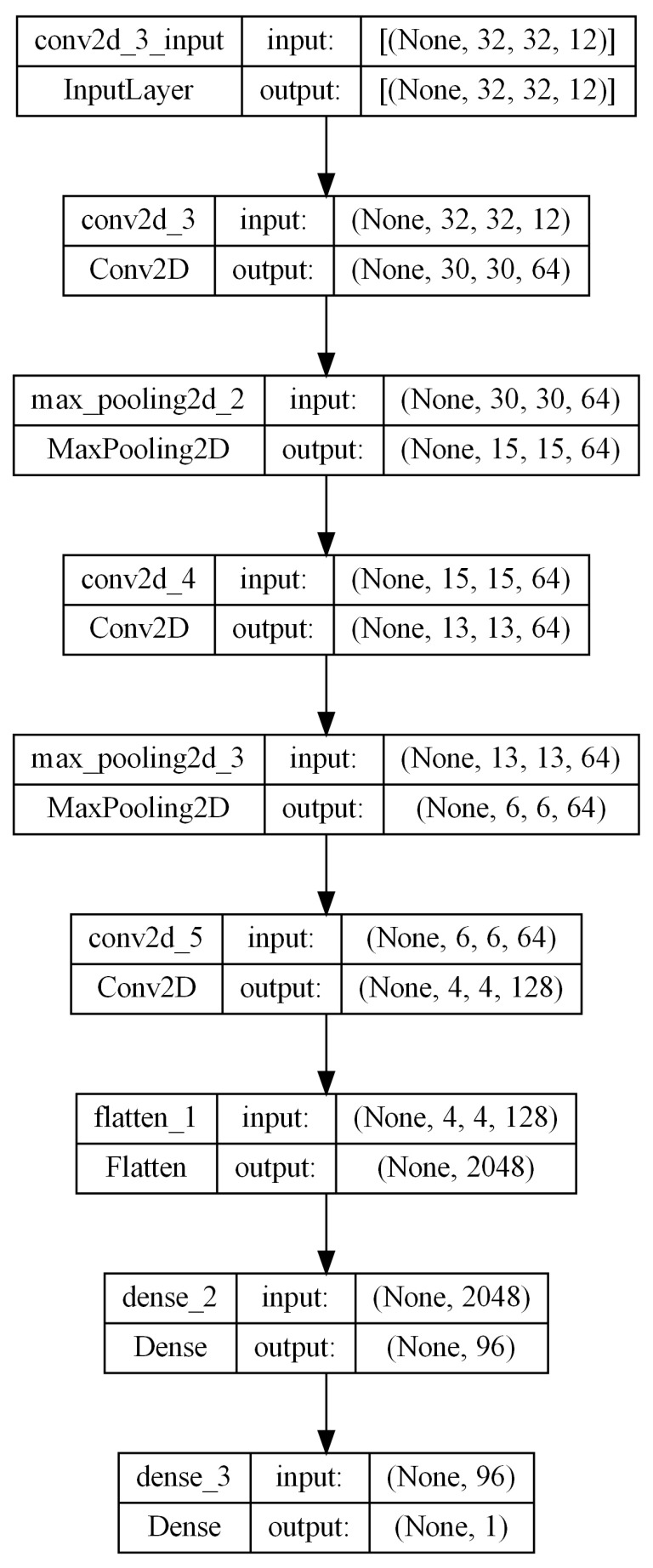
The implementation of the CNN models relied on the TensorFlow 2.0 framework [27] and KerasTuner [25]. An illustration depicting the architecture of the CNN model is provided in this figure.

**Figure 4 sensors-24-02432-f004:**
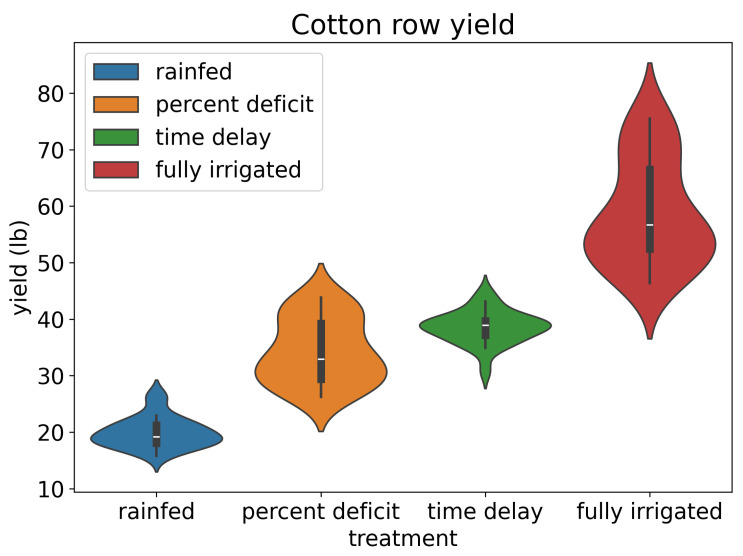
The violin plot illustrates the distribution of row cotton yield data across four different irrigation treatments. Each violin represents the probability density of yields within a specific treatment group, with wider sections indicating higher density regions.

**Figure 5 sensors-24-02432-f005:**
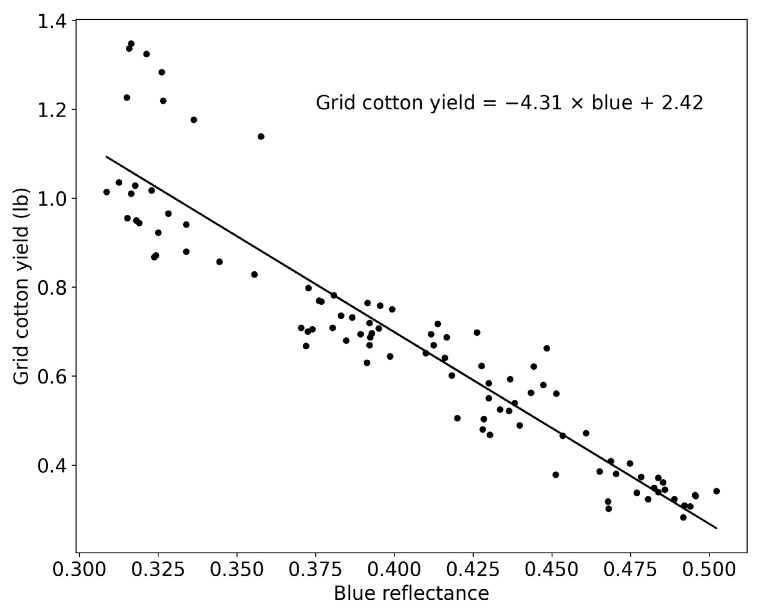
The regression analysis of blue reflectance and grid cotton yield.

**Figure 6 sensors-24-02432-f006:**
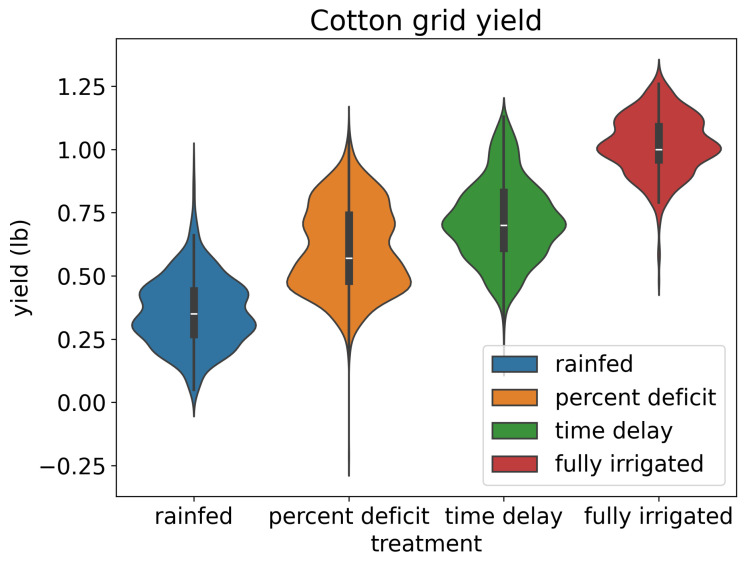
The violin plot illustrates the distribution of grid cotton yield data across four different irrigation treatments. Each violin represents the probability density of yields within a specific treatment group, with wider sections indicating higher density regions.

**Figure 7 sensors-24-02432-f007:**
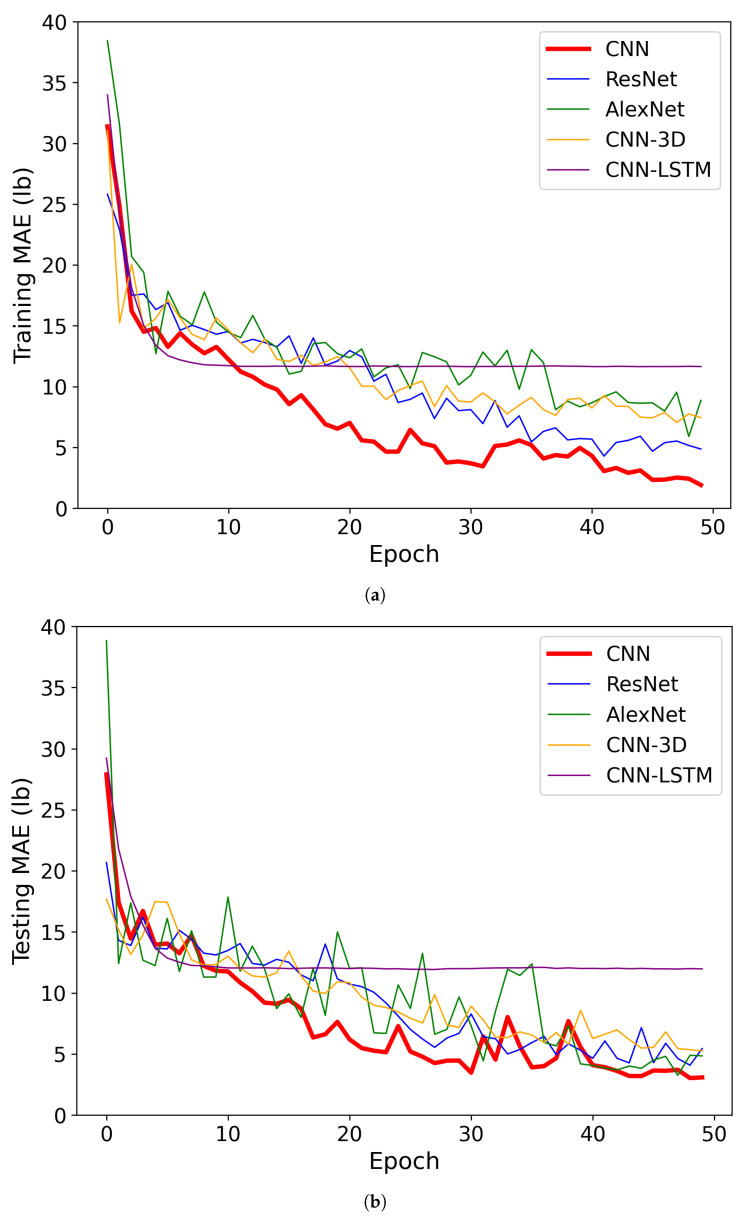
The training and testing performance of the CNN models with the cotton row image dataset. (**a**) Training performance at the cotton row level; (**b**) testing performance at the cotton row level.

**Figure 8 sensors-24-02432-f008:**
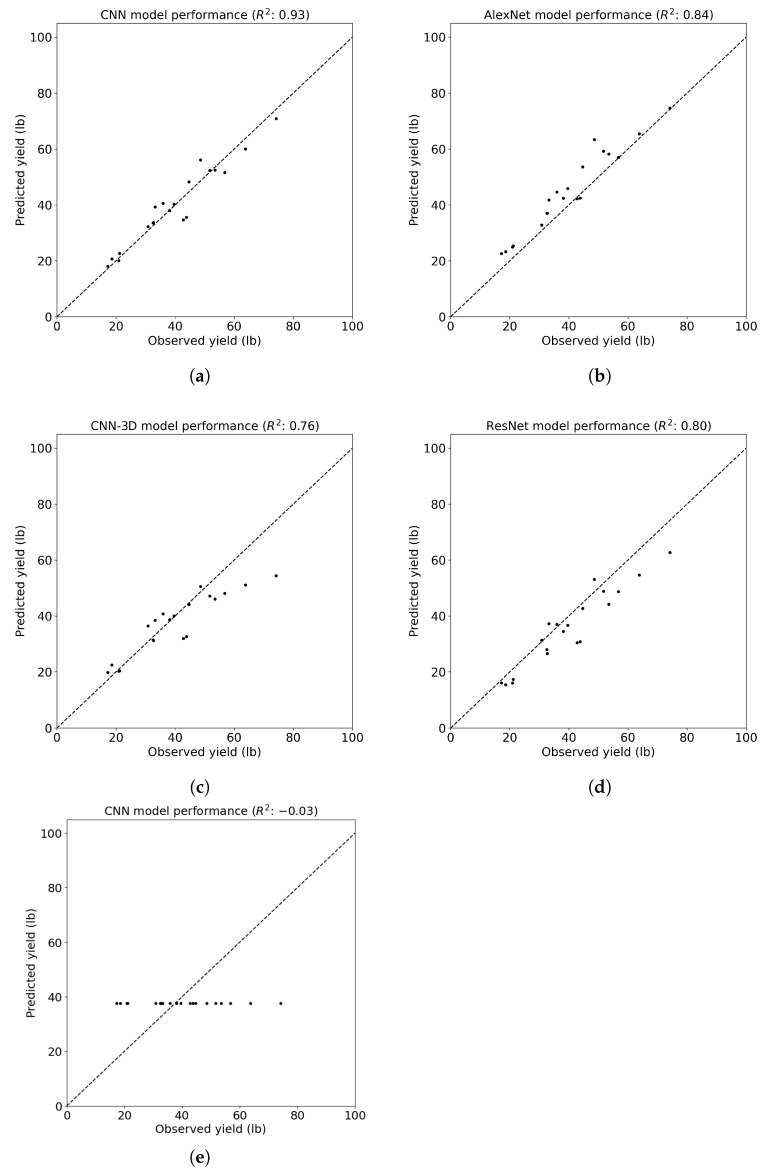
The $R^{2}$ of the CNN models for cotton yield prediction at the cotton row level. (**a**) The proposed CNN model; (**b**) the AlexNet model; (**c**) the CNN-3D model; (**d**) the ResNet model; (**e**) the CNN-LSTM model.

**Figure 9 sensors-24-02432-f009:**
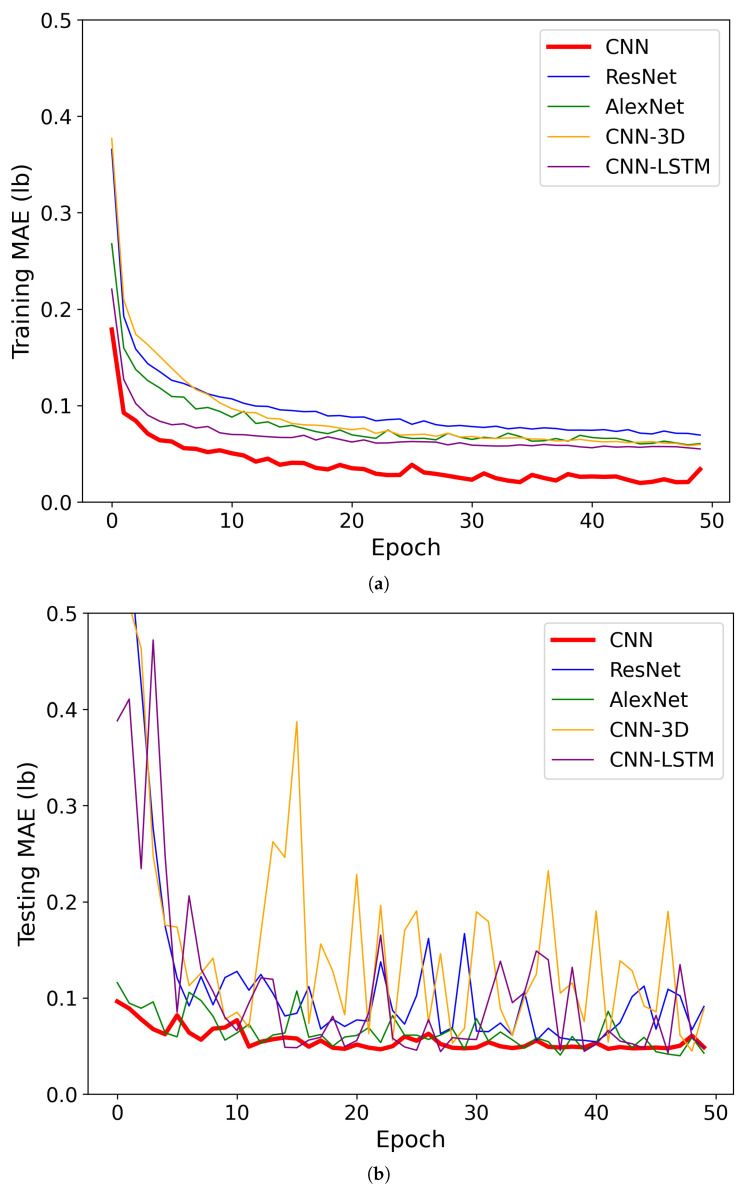
The training and testing performance of the CNN models with the cotton grid image dataset. (**a**) Training performance at the cotton grid level; (**b**) testing performance at the cotton grid level.

**Figure 10 sensors-24-02432-f010:**
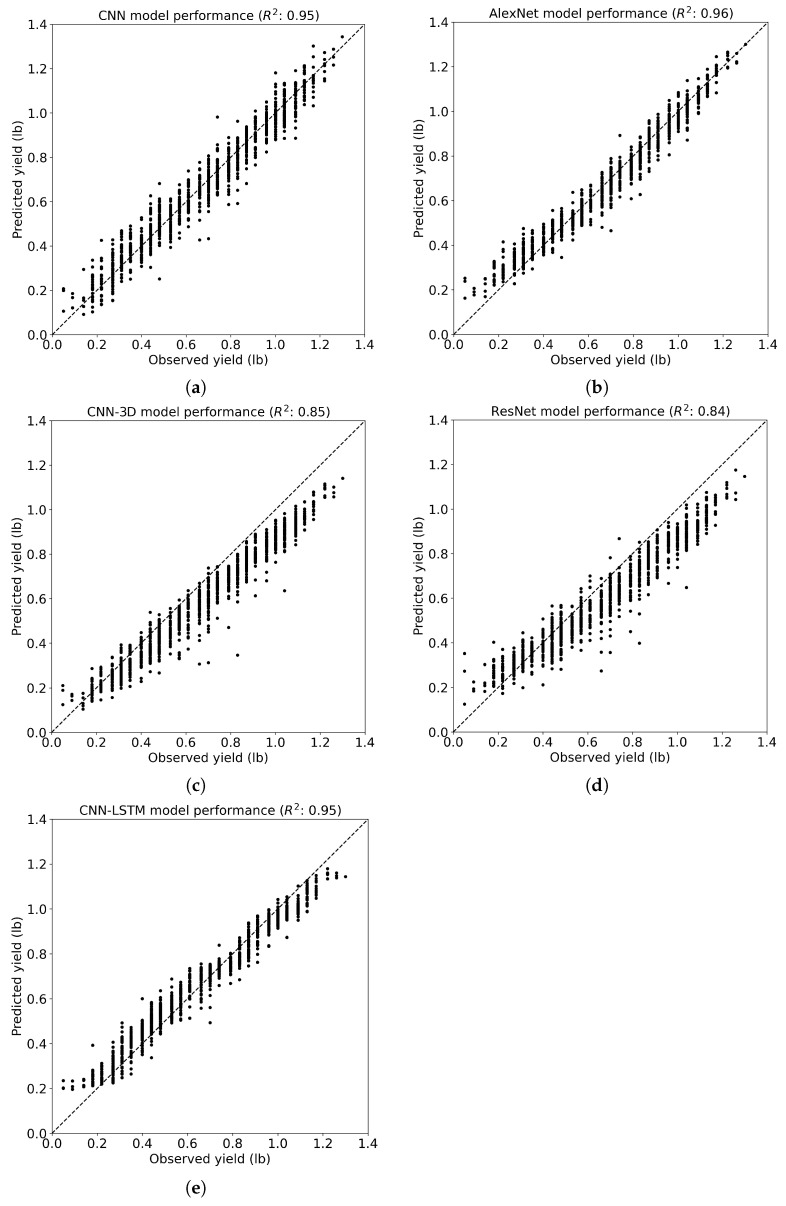
The $R^{2}$ of the CNN models for cotton yield prediction at the cotton grid level. (**a**) The proposed CNN model; (**b**) the AlexNet model; (**c**) the CNN-3D model; (**d**) the ResNet model; (**e**) the CNN-LSTM model.

**Table 1 sensors-24-02432-t001:** Summary of the cotton yield at different field scale and irrigation treatments.

Data	Treatment	Mean (lb)	Std (lb)	Count
Grid yield	Rainfed	0.36	0.14	1344
	Percent deficit	0.61	0.17	1344
	Time delay	0.72	0.16	1344
	Fully irrigated	1.02	0.12	1344
Row yield	Rainfed	19.92	2.66	24
	Percent deficit	34.26	5.72	24
	Time delay	38.57	2.89	24
	Fully irrigated	58.68	9.31	24

**Table 2 sensors-24-02432-t002:** The performance of CNN models at the row level.

CNN Models	MAE (lb)	MAPE (%)	$R^{2}$
AlexNet	4.84	14.4	0.84
ResNet	5.44	13.53	0.80
CNN-3D	5.25	12.08	0.76
CNN-LSTM	11.97	35.07	−0.03
Proposed CNN	3.08	7.76	0.93

**Table 3 sensors-24-02432-t003:** The performance of CNN models at the grid level.

CNN Models	MAE (lb)	MAPE (%)	$R^{2}$
AlexNet	0.05	10.08	0.96
ResNet	0.09	15.61	0.84
CNN-3D	0.09	14.17	0.85
CNN-LSTM	0.05	10.46	0.95
Proposed CNN	0.05	10.00	0.95

## Data Availability

The dataset will be available upon request.

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
