# Peer review of "In-Season Cotton Yield Prediction with Scale-Aware Convolutional Neural Network Models and Unmanned Aerial Vehicle RGB Imagery"

_sensors, 2024, doi:10.3390/s24082432_

Round 1

Reviewer 1 Report

Comments and Suggestions for Authors

After a thorough review of the submitted manuscript, I find that it lacks a significant novel scientific contribution, primarily because the proposed CNN-LSTM network model does not introduce new insights or advancements beyond the current state of research in this domain. Additionally, the similarity rate of 26% raises concerns about the originality of the content, with a particularly high concentration (12%) of similarity from a single source. Such levels of similarity suggest a reliance on existing literature that may not meet the journal's standards for original research contributions. Given these critical issues, my recommendation is to decline this submission. Future submissions should aim for a clear demonstration of novelty and adherence to originality standards to align with the journal's publication criteria.

Moreover, the comparison of results is limited to baseline deep learning models. For a comprehensive evaluation of the proposed method's efficacy, it is imperative that comparisons are also made with state-of-the-art models. This would provide a clearer understanding of the proposed method's standing in the current research landscape and its potential contribution to advancing the field.

Given these concerns, particularly the lack of a novel scientific contribution and the high similarity rate, I recommend that the paper be declined for publication in the Sensors Journal. Future submissions should aim to address these critical issues, focusing on originality, innovation, and a broader comparative analysis with the latest advancements in the field.

Author Response

Thanks for your comments.

  1. We have carefully compared with our two papers. The similarity mainly comes from descriptive or functional words, such as the experimental field description, the last paragraph of the Introduction, etc. We have modified those words, marked as red in the draft. The similarity has nothing to do with the CNN methods and research topics. In our previous paper, a simple fixed CNN model structure was applied for cotton water stress classification task. In this paper, the CNN models combined with a KerasTuner method (Page 3, from line 99 to line 122) was applied for cotton yield prediction as regression models.

  1. Regarding the deep learning models, we built a reproducible, customized CNN framework for cotton yield prediction task. Researchers can reuse this framework to compare with their own methods in the future. This framework can be scalable, people can add more advanced CNN methods if researchers are willing to share their code or dataset. More complex deep learning models does not always mean better performance when switched to a new dataset or scenario, it can easily lead to generalization issues for agricultural dataset. People may need more time effort to do parameter tuning for the others’ advanced models. For our cotton yield data, all the CNN models mentioned in this paper, such as CNN, CNN-LSTM, CNN-3D, AlexNet, ResNet, were initially set up with more complex structures but does not lead to better performance. It can be proved by the KerasTuner method in our code. With our methods proposed in the paper, people can spend less time on hand parameter tuning effort and generate a relatively high cotton yield prediction performance. At the field scales we mentioned in the paper, the current state-of-art performance is around 5 – 10 % (mean absolute percentage error). So, our models should be able to be used or compared as baseline models.

  1. For the contribution, briefly speaking, 1. we applied KerasTuner method to automatically search the best CNN model structures for cotton yield prediction. 2. we built a reproducible, customized CNN framework for cotton yield prediction task. Researchers can reuse this framework to compare with their own methods in the future. This framework can be scalable, people can add more advanced CNN methods if researchers are willing to share their code or dataset. 3. Currently we see few people discussing the cotton yield variability at such high scale (1 square meter), our current methods may give some hints for future research. 4. Our dataset is well organized, and we are willing to share it with others.

Reviewer 2 Report

Comments and Suggestions for Authors

In this article, a new approach to cotton yield prediction is proposed by leveraging the synergy between Unmanned Aerial Vehicles (UAVs) and scale-aware convolutional neural networks (CNN). The manuscript shows a  clear and well-organized presentation. However, a major revision is required before the article can be published.

1) The similarity of the article to the previously published “Classification of cotton water stress using convolutional neural networks and UAV-based RGB imagery”  article should be reduced.

2) The differences and advantages of this article compared to the authors' previous article (“Classification of cotton water stress using convolutional neural networks and UAV-based RGB imagery”) should be clearly stated.

3) The novelty of the proposed methodology should be emphasized.

4) The conclusions section should be improved and more detailed.

5) The resolution quality of Figure 2 should be improved.

6) The proposed CNN model in Table 3 should be given at the bottom.

7) 2.1. The title of The Study Site and Yield Data Collection should not start directly with a figure. Figure 1 can be given between or at the end of the texts.

Comments on the Quality of English Language

In this article, a new approach to cotton yield prediction is proposed by leveraging the synergy between Unmanned Aerial Vehicles (UAVs) and scale-aware convolutional neural networks (CNN). The manuscript shows a  clear and well-organized presentation. However, a major revision is required before the article can be published.

1) The similarity of the article to the previously published “Classification of cotton water stress using convolutional neural networks and UAV-based RGB imagery”  article should be reduced.

2) The differences and advantages of this article compared to the authors' previous article (“Classification of cotton water stress using convolutional neural networks and UAV-based RGB imagery”) should be clearly stated.

3) The novelty of the proposed methodology should be emphasized.

4) The conclusions section should be improved and more detailed.

5) The resolution quality of Figure 2 should be improved.

6) The proposed CNN model in Table 3 should be given at the bottom.

7) 2.1. The title of The Study Site and Yield Data Collection should not start directly with a figure. Figure 1 can be given between or at the end of the texts.

Author Response

In this article, a new approach to cotton yield prediction is proposed by leveraging the synergy between Unmanned Aerial Vehicles (UAVs) and scale-aware convolutional neural networks (CNN). The manuscript shows a  clear and well-organized presentation. However, a major revision is required before the article can be published.

  1. The similarity of the article to the previously published “Classification of cotton water stress using convolutional neural networks and UAV-based RGB imagery” article should be reduced.

Response- We have modified the descriptive or functional words to reduce the similarity. The modification was marked as red in the paper.

  1. The differences and advantages of this article compared to the authors' previous article (“Classification of cotton water stress using convolutional neural networks and UAV-based RGB imagery”) should be clearly stated.

Response- We have added the discussion in the article. The main difference is that the previous CNN was a fixed structure for classification task. We applied KerasTuner for the current CNN models so we can do parameter tunings automatically. The current CNN models are for regression tasks. See more details in the paper, Page 3, from line 99 to line 122.

  1. The novelty of the proposed methodology should be emphasized.

Response- Thanks for the comment. In page 3, from line 99 to line 132, we discussed the innovation and contribution of our work.

Briefly speaking, 1. we applied KerasTuner method to automatically search the best CNN model structures for cotton yield prediction. 2. we built a reproducible, customized CNN framework for cotton yield prediction task. Researchers can reuse this framework to compare with their own methods in the future. This framework can be scalable, we can add more advanced CNN methods if researchers are willing to share their code or dataset. 3. Currently we see few people discussing the cotton yield variability at such high scale (1 square meter), our current methods may give some hints for future research. 4. Our dataset is well organized, and we are willing to share it with others.  

  1. The conclusions section should be improved and more detailed.

Response- We have modified the conclusion.

  1. The resolution quality of Figure 2 should be improved.

Response- Thanks for pointing it out. We are saving it as 300 dpi. However, Figure 2 is the best image quality we can get because of the UAV flight height (90 meters) and camera specs. Each image in figure 2 is only about 1 m2.

  1. The proposed CNN model in Table 3 should be given at the bottom.

Response- Thanks for the comment. The proposed CNN model is given at the bottom now.

  1. 1. The title of The Study Site and Yield Data Collection should not start directly with a figure. Figure 1 can be given between or at the end of the texts

Response- We have moved the figure to the end of the texts in page 4.

Round 2

Reviewer 1 Report

Comments and Suggestions for Authors

The revised version of the paper presents a comprehensive study on "In-season cotton yield prediction with scale-aware CNN models and UAV RGB imagery." The authors have satisfactorily addressed the concerns raised in the previous review, demonstrating a commitment to enhancing the quality and clarity of their work. This review will discuss the positive aspects and limitations of the revised manuscript, leading to a final recommendation.

Positive Aspects

Practicability: The most commendable aspect of this paper is its practical applicability. The study introduces a novel approach to cotton yield prediction, leveraging the synergy between Unmanned Aerial Vehicles (UAVs) and scale-aware convolutional neural networks (CNN). This approach is not only innovative but also directly applicable to real-world agricultural scenarios, offering a potent strategy for advancing precision agriculture. By providing growers with a powerful tool to optimize cultivation practices and enhance overall cotton productivity, the paper makes a significant contribution to the field.

Comprehensive Evaluation: The authors have conducted a thorough evaluation of their proposed models, demonstrating their superiority over conventional CNN architectures such as AlexNet, CNN-3D, CNN-LSTM, and ResNet. The use of KerasTuner to dynamically search the best CNN model structure for cotton yield prediction is a notable methodological strength, ensuring that the findings are robust and reliable.

Research Reproducibility: Another positive aspect is the commitment to research reproducibility. The authors have made the code and dataset available on GitHub, facilitating further research and verification of their results by the scientific community.

Limitations

Low Novelty in Proposed Models: Despite the practical implications of the study, a limitation lies in the low novelty of the proposed models. The convolutional neural network (CNN) framework, while effectively employed in this study, does not introduce groundbreaking changes to existing models or techniques. The innovation primarily resides in the application and optimization of these models rather than in the development of new architectural principles or learning algorithms. This aspect may limit the paper's impact on advancing the theoretical understanding of CNN applications in precision agriculture.

In light of the above considerations, it is my opinion that the paper's strengths, particularly its practical applicability and thorough evaluation, outweigh its limitations regarding the novelty of the proposed models. Consequently, I deem the manuscript acceptable for publication. It is recommended, however, that future work focus on developing more innovative modeling techniques to further advance the field.

Reviewer 2 Report

Comments and Suggestions for Authors

Authors revised the manuscript well and replied to all of my comments satisfactorily. 

Comments on the Quality of English Language

Authors revised the manuscript well and replied to all of my comments satisfactorily.